

# Comparative efficacy of different exercise interventions in patients with ankylosing spondylitis: a systematic review and network meta-analysis

Lingkui Kong[1,*], Chuanwen Yu[1,*], Chaoxin Wang[1] and Zhanpeng Meng[2]

[1] College of Physical Education and Health, Heze University, Heze, China
[2] College of Physical Education, Shandong Normal University, Jinan, Shandong, China
[*] These authors contributed equally to this work.

## ABSTRACT

**Background**. Exercise interventions have been widely applied as an adjunctive treatment for ankylosing spondylitis (AS), effectively alleviating pain, improving function, and enhancing quality of life. However, the efficacy of different exercise modalities remains inconclusive. This study aims to systematically compare the effects of various exercise interventions on clinical outcomes in AS patients using a network meta-analysis (NMA) to determine the optimal exercise regimen.

**Methods**. A comprehensive search was performed across six databases to identify studies assessing the impact of exercise interventions on AS. The primary outcomes analyzed included the Bath Ankylosing Spondylitis Disease Activity Index (BASDAI), Bath Ankylosing Spondylitis Functional Index (BASFI), Bath Ankylosing Spondylitis Metrology Index (BASMI), and Ankylosing Spondylitis Quality of Life (ASQoL). A network meta-analysis was conducted using the frequentist approach in STATA 18.0, while effect publication bias analysis used Review Manager 5.4.

**Results**. Forty-eight studies involving 3,140 participants were included, published between 2002 and 2024. Compared to the control group, all exercise interventions demonstrated varying degrees of benefit in improving BASDAI, BASFI, BASMI, and ASQoL scores. For BASDAI, the Aquatic Stretching Exercise (ASE) (−1.42, 95% CI [−2.51 to −0.33]), Land Aerobic Exercise (LAE) (−0.94, 95% CI [−1.41 to −0.47]), and Land Stretching Exercise (LSE) (−0.49, 95% CI [−0.94 to −0.04]) exhibited significant symptom relief, with ASE ranking highest (surface under the cumulative ranking curve (SUCRA) = 85.5). For BASFI, statistically significant improvements were observed with the Aquatic Aerobic Exercise (AAE) (−0.90, 95% CI [−1.50 to −0.29]), ASE (−1.74, 95% CI [−2.45 to −1.04]), LAE (−0.74, 95% CI [−1.05 to −0.43]), LSE (−0.54, 95% CI [−0.81 to −0.27]), and Muscle Exercise (ME) (−0.48, 95% CI [−0.83 to −0.13]). ASE had the highest SUCRA ranking (99.6). For BASMI, ASE (−1.06, 95% CI [−2.10 to −0.02]), LAE (−0.51, 95% CI [−1.01 to −0.01]), and the China Health Campaign (CHC) (−1.15, 95% CI [−2.18 to −0.13]) showed significant improvements, with CHC ranking highest (SUCRA = 78.7). For ASQoL, ASE (−3.67, 95% CI [−6.17 to −1.18]) and LAE (−2.64, 95% CI [−4.50 to −0.79]) demonstrated statistical significance, with ASE achieving the highest ranking (SUCRA = 88.4).

Corresponding author
Chuanwen Yu,
yuchuanwen@hezeu.edu.cn

**Conclusion**. This NMA systematically evaluated the effectiveness of different exercise interventions on clinical outcomes in AS patients. All exercise modalities provided varying degrees of benefit compared to the control group. ASE exhibited the most significant improvements in BASDAI, BASFI, and ASQoL, suggesting its superiority as an intervention. Additionally, CHC demonstrated the most significant potential for BASMI improvement. ASE is the most effective exercise modality for symptom relief, functional enhancement, and quality-of-life improvement, warranting further promotion in clinical practice. **Registration** PROSPERO (No. CRD42025639115).

# INTRODUCTION

Ankylosing spondylitis (AS) is a chronic progressive inflammatory disease primarily affecting the sacroiliac joints and the spine. It is a representative subtype of spondylo arthritis. The clinical features of AS include chronic low back pain, morning stiffness, and restricted spinal mobility. If inadequately controlled, the disease may progress to spinal ankylosis, functional impairment, and even severe disability. The global prevalence of AS is approximately 0.48% in Northern Europe and 0.20% in Asia (*Boel et al., 2022*). Reports indicate that approximately 30% of AS patients develop spinal bony fusion within 10 years of disease onset, with some patients also experiencing involvement of the cardiovascular system, eyes, or lungs, significantly increasing the risk of mortality (*Poddubnyy et al., 2024*; *Shao et al., 2021*). Studies have shown that the all-cause mortality rate in AS patients is 60% higher than that of the general population, primarily due to cardiovascular complications and respiratory dysfunction resulting from spinal fractures (*Bittar et al., 2024*). Furthermore, AS predominantly affects young and middle-aged adults, often leading to long-term reductions in work capacity and substantial economic burdens, with an estimated annual productivity loss of $18,952 per patient (*Magrey et al., 2024*; *Narayanan et al., 2013*). Therefore, the effective management of AS is of significant clinical importance.

Currently, the treatment of AS primarily relies on pharmacological interventions, including nonsteroidal anti-inflammatory drugs (NSAIDs), glucocorticoids, conventional synthetic disease-modifying antirheumatic drugs (csDMARDs), and biologic agents such as TNF-$\alpha$ inhibitors and IL-17 inhibitors. These medications have demonstrated varying degrees of efficacy in alleviating inflammation and relieving pain, with biological agents, in particular, representing a breakthrough in controlling disease activity in recent years. However, NSAIDs, despite being the first-line treatment, can only provide temporary pain relief and do not effectively slow the progression of structural damage (*Kim et al., 2024*). Additionally, prolonged NSAID use is associated with severe adverse effects; studies suggest that approximately 34% of patients develop gastric mucosal injury, while 9.8% experience

acute kidney injury (*Kroon et al., 2015*). Biologic agents, particularly TNF-*α* inhibitors, have been widely adopted to address these limitations due to their proven efficacy in inflammation control. However, they are costly and carry an increased risk of infections (*Garcia-Montoya & Emery, 2021*). Despite these advancements, approximately 40% to 50% of AS patients exhibit inadequate responses to existing therapies, failing to achieve adequate disease control and potentially accelerating spinal deformities (*Siderius et al., 2024*). This underscores the limitations of pharmacological treatment in improving structural damage, postural deformities, and functional impairments. Moreover, the long-term use of these medications is associated with immunosuppression, infection risks, and high treatment costs. Consequently, pharmacological therapy alone is insufficient to comprehensively address the multifaceted needs of AS patients, particularly in terms of quality of life and functional recovery.

In 2022, the Assessment of SpondyloArthritis International Society (ASAS) and the European League Against Rheumatism (EULAR) jointly issued updated guidelines, which, for the first time, elevated exercise therapy from an adjunctive treatment to a cornerstone intervention in the management of AS (*Ramiro et al., 2023*). Accumulating evidence supports the biological and functional benefits of exercise therapy in AS. On the molecular level, physical activity has been shown to reduce pro-inflammatory cytokines such as IL-17, a key component of the IL-17/IL-23 axis, thereby contributing to the suppression of inflammatory responses (*Golzari et al., 2010*). Meanwhile, at the functional level, it alleviates pain, improves physical function and spinal mobility, and enhances quality of life in patients with axSpA (*Wang et al., 2024*). Clinical studies have demonstrated that exercise interventions can significantly reduce pain, improve spinal mobility, and modulate inflammation. Various modalities—including individualized functional training (*Boudjani et al., 2023*; *Luo et al., 2024*), postural correction (*Basakci Calik et al., 2018*; *Sveaas et al., 2020*), breathing exercises, and aquatic-based programs (*Gandomi et al., 2022*; *Dundar et al., 2014*)—have shown positive effects on physical function and quality of life in AS patients. Furthermore, research has begun to elucidate the mechanistic differences among exercise modalities: aerobic exercise reduces disease activity and improves cardiopulmonary function, potentially through the modulation of pro-inflammatory cytokines such as TNF-*α* and IL-6 (*Acharya et al., 2024*; *Wang et al., 2022*); strength training improves muscle strength and functional mobility in AS patients. For example, *De Souza et al. (2017)* showed that a 16-week program of Swiss-ball progressive resistance exercises significantly increased one-repetition maximum across several exercises and enhanced walking ability in AS patients; other types of flexibility training—like Pilates (*Altan et al., 2012*) and stretching exercises (*Gandomi et al., 2022*)—have also been shown to improve respiratory coordination and decrease stiffness in related populations, although more direct evidence for AS is still developing. Notably, due to its low-impact nature and high coordination demands, aquatic exercise is particularly beneficial for patients with limited mobility or severe pain, as it improves physical function without exacerbating joint stress (*Zão & Cantista, 2017*; *Gunay, Keser & Bicer, 2018*).

However, despite these promising findings, the question of which specific exercise modality is most effective for AS management remains unresolved. Previous systematic

reviews have attempted to address this topic. For example, *Boudjani et al. (2023)* conducted a meta-analysis examining how exercise affects pain and function in patients with ankylosing spondylitis. Although their study included various exercise programs, it did not provide detailed comparisons between specific modalities (such as aerobic *versus* flexibility training), nor did it rank interventions against each other. Similarly, *Lane et al. (2022)* studied the psychological effects of group- and home-based exercise programs, like anxiety and depression, but did not assess physical or functional outcomes, nor did they include aquatic or mind–body exercises like Pilates or tai chi. Additionally, neither study used a network meta-analysis approach to compare the relative effectiveness of different intervention types. Furthermore, observational studies have shown that higher physical activity levels are linked to better physical function, lower disease activity, improved quality of life, and more favorable psychological outcomes in patients with AS (*Demontis et al., 2016*; *Aytekin et al., 2012*). These limitations and observational findings highlight the need for a more comprehensive synthesis that incorporates various exercise types and provides clinically actionable comparisons.

The current study uses a network meta-analysis to assess and rank the relative effectiveness of various exercise interventions in AS. Unlike traditional pairwise meta-analysis, this method allows for the comparison of multiple interventions at once by combining both direct and indirect evidence. Specifically, this study aims to: (1) systematically compare the effects of different exercise methods on key clinical outcomes in AS patients, including Bath Ankylosing Spondylitis Disease Activity Index (BASDAI), Bath Ankylosing Spondylitis Functional Index (BASFI), Bath Ankylosing Spondylitis Metrology Index (BASMI), and Ankylosing Spondylitis Quality of Life (ASQoL); (2) determine which interventions are most effective across different symptom areas; and (3) provide evidence-based guidance to help tailor exercise plans in clinical practice. By addressing these research questions, this study seeks to build on existing evidence and offer new insights into the most effective exercise therapy design for patients with AS.

## METHODS

This systematic review follows the PRISMA for network meta-analyses (NMA) guidelines for network meta-analyses (*Hutton et al., 2015*) and complies with the methodology outlined in the Cochrane Handbook for Systematic Reviews (*Higgins et al., 2024*). The meta-analysis has been officially registered with PROSPERO under the identifier CRD42025639115.

### Search strategy and information sources

A comprehensive literature search was performed in six significant databases: CNKI, Web of Science (WOS), Cochrane Library, Embase, Scopus, and PubMed. The search spanned from the inception of each database to March 5, 2025, using a combination of controlled vocabulary and free-text terms. Keywords included ''Ankylosing Spondylitis,'' ''Spondylarthritis,'' ''AS,'' ''Exercise,'' ''Physical Activity,'' ''Sport,'' and ''Training.'' The whole search strategy is detailed in Appendix S1. Independently, two trained researchers, L.K. K. and C.W. Y., conducted the literature search and screened the retrieved studies

based on predefined inclusion and exclusion criteria. Any disagreements between the two reviewers were resolved through discussion. If consensus could not be reached, a third reviewer, C.X.W., was consulted to make the final decision and ensure selection accuracy and consistency.

## Study selection

In this study, we classified the included exercise interventions into six distinct categories based on modality, environment (land *vs.* aquatic), and core training focus (aerobic, stretching, muscle exercise, or China health campaign). This classification reflects both physiological mechanisms and clinical application patterns described in existing guidelines and literature. Aquatic therapies are categorized separately because of their unique biomechanical and thermal features, which notably change joint loading and patient experience. The primary physiological goal of the exercise determines its category. For example, interventions aimed at improving endurance are classified as aerobic, while those focusing on joint range of motion are labeled as stretching. CHC interventions are distinctly grouped because they combined physical, respiratory, and psychological components specific to Eastern medical traditions.

Initially, all retrieved records were deduplicated using EndNote X9. Subsequently, two researchers (C.W. Y. and Z.P. M.) independently screened the titles and abstracts to identify studies that met the inclusion criteria. A full-text review was conducted to assess eligibility and quality for studies further initially deemed eligible. Any disagreements in the selection process were resolved through discussion, and if consensus could not be reached, a third researcher (C.X.W.) in the field was consulted for arbitration. The specific inclusion criteria were as follows: (a) The study design had to be either an RCT or an nRCT. (b) Participants had to be patients with a confirmed AS diagnosis. (c) The intervention had to involve one of the following exercise types: Land Aerobic Exercise (LAE) (*e.g.*, running, yoga), Land Stretching Exercise (LSE) (*e.g.*, stretching), Aquatic Aerobic Exercise (AAE) (*e.g.*, water yoga, swimming), Aquatic Stretching Exercise (ASE) (*e.g.*, water Pilates), China Health Campaign (CHC) (*e.g.*, Tai Chi, Baduanjin), or Muscle Exercise (ME) (*e.g.*, HIIT, respiratory muscle training). (d) The control group received no exercise intervention or underwent routine care, health education, or sham exercise training rather than active exercise therapy. (e) The study had to report at least one of the following outcome measures: BASDAI, BASFI, BASMI, or ASQoL score. The exclusion criteria were as follows: (a) Studies with erroneous or missing raw data that prevented valid data extraction. (b) Studies where the intervention group was not exclusively composed of AS patients, including other diseases or acute relapsing AS cases. (c) Studies without full-text availability, conference abstracts, review articles, or other secondary research. (d) Duplicate publications: If multiple studies originated from the same cohort with overlapping or redundant results, only the most comprehensive or most recently published study was included. (e) Studies involving exercise simulations or virtual interventions. For example, we excluded the study by *Salbaş & Karahan (2023)*.

## Data extraction

Two researchers (K.L.K and C.W.Y) independently extracted key information from the included studies. After initial extraction, they cross-checked each other's data to ensure accuracy and consistency. Any discrepancies were resolved through discussion, and if needed, a third researcher (C.X.W) provided arbitration. The finalized data were used for the systematic review and subsequent quantitative analysis. The extracted data included: (a) Study details: first author and publication year. (b) Participant characteristics: sample size, average age, and gender distribution. (c) Intervention details: type of exercise intervention, standard treatment description, and intervention duration. (d) Outcome measures: primary study results, including BASDAI, BASFI, BASMI, and ASQoL.

## Statistical analysis

This study utilized Stata 18.0 software for statistical processing and analysis. For a detailed description of the procedures used to perform network meta-analysis in Stata, readers are referred to *Shim et al. (2017)*, who provide a comprehensive methodological framework for conducting network meta-analyses using frequentist models. To assess the risk of bias, Cochrane's Risk of Bias tool for RCTs (*Higgins et al., 2011*) was applied independently by two reviewers (K.L.K and C.W.Y). Any discrepancies in the assessment were resolved through discussion with a third reviewer (C.X.W). Bias risk levels were classified according to Cochrane standards as "low risk", "unclear risk", or "high risk" and recorded accordingly. For continuous outcome variables, the mean deviation (MD) was used to estimate the pooled effect size, Transformation formulas for estimating the mean and standard deviation see Appendix S3. NMA was performed using a frequentist approach, combining direct and indirect comparison evidence. In the network structure diagram, each node represents an intervention, with its size proportional to the number of participants receiving that intervention. The thickness of the connecting lines indicates the number of studies that directly compared two interventions. Network consistency was examined using global and local inconsistency tests (loop inconsistency test). The NMA results were displayed in forest plots, where the zero line represents no effect (0), and results were interpreted based on effect sizes and 95% confidence intervals. To rank the effectiveness of interventions, we used surface under the cumulative ranking curve (SUCRA) values and mean rankings. SUCRA values (ranging from 0% to 100%) indicate the probability of an intervention ranking higher in effectiveness. A higher SUCRA value suggests a more effective intervention. Lastly, a funnel plot was used to evaluate publication bias across the included studies.

# RESULTS

## Basic features of the included literature

The PRISMA flow diagram outlines the study selection process (Fig. 1). We used two previous reviews to identify potentially eligible studies for inclusion (*Luo et al., 2024*; *Boudjani et al., 2023*). A total of 4,014 studies related to AS were identified through database searches. Based on predefined inclusion and exclusion criteria, the research

team conducted a rigorous screening process, ultimately including 48 eligible studies—comprising 41 RCTs and seven nRCTs—involving 3,140 diagnosed AS patients, with 1,213 males and 947 females. The basic characteristics of the included studies are provided in Table 1. The included studies examined seven different types of exercise interventions, specifically: four studies introduced AAE (*e.g.*, water walking, aquatic jogging, or water aerobics) in the experimental group (*Gandomi et al., 2022*; *Dundar et al., 2014*; *Kjeken et al., 2013*; *Karapolat et al., 2009*). Four studies implemented ASE (*Gandomi et al., 2022*; *Gunay, Keser & Bicer, 2018*; *Gurcay et al., 2008*; *Ciprian et al., 2013*). Twenty-five studies used LAE (*e.g.*, walking, jogging, or aerobic workouts) (*Basakci Calik et al., 2018*; *Sveaas et al., 2020*; *Dundar et al., 2014*; *Acharya et al., 2024*; *Wang et al., 2022*; *Gunay, Keser & Bicer, 2018*; *Karapolat et al., 2009*; *Gurcay et al., 2008*; *Basakci Calik et al., 2021*; *Karahan et al., 2016*; *Demontis et al., 2016*; *Karapolat et al., 2008*; *Masiero et al., 2011*; *So et al., 2012*; *Analay et al., 2003*; *Aytekin et al., 2012*; *Cagliyan et al., 2007*; *Günendi et al., 2010*; *Hsieh et al., 2014*; *Jennings et al., 2015*; *Karahan et al., 2016*; *Niedermann et al., 2013*; *Roşu et al., 2014*; *Rosu & Ancuta, 2015*; *Silva, Andrade & Vilar, 2012*). Twenty-three studies applied LSE (*García et al., 2015*; *Aksoy et al., 2017*; *Singh et al., 2023*; *De Souza et al., 2017*; *Li et al., 2017*; *Karapolat et al., 2009*; *Altan et al., 2012*; *Analay et al., 2003*; *Basakci Calik et al., 2021*; *Günendi et al., 2010*; *Hsieh et al., 2014*; *Jennings et al., 2015*; *Karapolat et al., 2009*; *Kasapoglu Aksoy et al., 2017*; *Masiero et al., 2014*; *Rodriguez-Lozano et al., 2013*; *Roşu et al., 2014*; *Rosu & Ancuta, 2015*; *Silva, Andrade & Vilar, 2012*; *De Souza et al., 2017*; *Sweeney, Taylor & Calin, 2002*; *Acharya et al., 2024*; *Cagliyan et al., 2007*). Five studies examined CHC (*e.g.*, Tai Chi, Baduanjin, and Qigong) (*Chen, 2014*; *Qu et al., 2020*; *Lee et al., 2008*; *Levitova et al., 2016*; *Xie et al., 2019*). Five studies investigated ME (*e.g.*, bodyweight training, resistance band exercises, or machine-based resistance training) (*Basakci Calik et al., 2018*; *So et al., 2012*; *Sveaas et al., 2014*; *Durmus et al., 2009*; *Widberg, Karimi & Hafström, 2009*).

## Network geometry

We assessed the effectiveness of six distinct exercise interventions for patients with ankylosing spondylitis. Figure 2 visually represents the direct comparisons among these interventions, where each node corresponds to a specific intervention. The size of each node reflects the sample size of that intervention, while the thickness of the connecting lines indicates the number of direct comparisons conducted between the two interventions. This network diagram offers a clear perspective on the relationships among interventions and the comprehensiveness of the evidence network. Figure 2A illustrates the network of different exercise interventions in relation to BASDAI. The analysis covers six intervention types: LAE, LSE, AAE, ASE, ME, and CHC. The largest nodes, corresponding to conventional treatment (CG), LAE, and LSE, indicate that these interventions had the most participants. The thickest line connecting CG and LAE suggests this pair was the most frequently compared indirect studies. The network is relatively well-balanced, with most interventions directly compared to the control group and some indirect comparison pathways present. Figure 2B depicts the network of exercise interventions concerning BASFI. Again, six interventions are included, with CG, LAE, and LSE having the largest sample sizes. The thicker connecting lines indicated that direct comparisons were particularly frequent
**Table 1  Study characteristics.**

| Study | Participants (exercise *vs.* control) | | | | Interventions | | Comparator | Outcomes (BASDAI, BASFI, BASMI, ASQoL) |
|---|---|---|---|---|---|---|---|---|
| | Methodology | Sample size | Age | Female/male | Type | Duration (wk) | | |
| García et al. (2015) | RCT | 15 *vs.* 15 | 43.8 ± 9.1 *vs.* 50 ± 13 | 16/14 | Strengthening, balance flexibility training | 8 | Conventional therapy | BASFI |
| Fernández-de Las-Peñas et al. (2005) | RCT | 17 *vs.* 14 | 46.58 ± 11.94 *vs.* 42.85 ± 11.07 | 12 *vs.* 19 | Running | 12 | Conventional therapy | BASDAI, BASFI, BASMI |
| Karahan et al. (2016) | RCT | 28 *vs.* 29 | 36.1 ± 12.4 *vs.* 36.6 ± 11.3 | 45 *vs.* 12 | Exergame | 8 | Conventional therapy | BASDAI, BASFI, ASQoL |
| Gandomi et al. (2022) | RCT | 14 *vs.* 14 | 39.21 ± 10.25 *vs.* 38.07 ± 8.69 | 28 *vs.* 0 | Aqua Pilates *vs.* Aqua Stretch | 6 | routine drug treatment | BASFI, ASQoL |
| Aksoy et al. (2017) | RCT | 20 *vs.* 21 | 37.95 ± 9.84 *vs.* 37.47 ± 11.09 | 32 *vs.* 9 | Stretch | 12 | Conventional therapy | BASDAI, BASFI, BASMI |
| Singh et al. (2023) | RCT | 57 *vs.* 52 | 34.42 ± 9.39 *vs.* 35.09 ± 9.86 | 91 *vs.* 18 | Yoga | 12 | Conventional therapy | BASDAI, BASFI, ASQoL |
| De Souza et al. (2017) | RCT | 27 *vs.* 28 | 45 ± 9.8 *vs.* 43.8 ± 10.2 | 41 *vs.* 14 | Swiss balls | 8/16 | Conventional therapy | BASDAI, BASFI, BASMI |
| Li et al. (2017) | RCT | 54 *vs.* 40 | 35.7 ± 8.7 *vs.* 36.5 ± 9.6 | 67 *vs.* 27 | strength, balance | 4/8/12 | Conventional therapy | BASDAI, BASFI |
| Chen (2014) | RCT | 54 *vs.* 40 | 35.7 ± 8.7 *vs.* 36.5 ± 9.6 | 67 *vs.* 27 | Tai Chi | 8 | Conventional therapy | BASDAI, BASFI |
| Qu et al. (2020) | RCT | 40 *vs.* 40 | 37.24 ± 14.13 *vs.* 40.38 ± 10.67 | 78 *vs.* 2 | Tai Chi | 4/8/12/16/32 | Conventional therapy | BASDAI, BASFI, BASMI |
| Basakci Calik et al. (2018) | RCT | 16 *vs.* 16 | 35.62 ± 8.18 *vs.* 39.12 ± 12.26 | 9/7 *vs.* 9/7 | Inspiratory muscle training | 8 | Conventional therapy | BASDAI, BASFI, BASMI |
| Demontis et al. (2016) | nRCT | 20 *vs.* 22 | 49.30 ± 11.33 *vs.* 45 ± 8.45 | 36/10 | supervised training and home-based rehabilitation program | 7/28 | No intervention | BASDAI, BASFI, BASMI |
| Dundar et al. (2014) | RCT | 35 *vs.* 34 | 42.3 ± 11.3 *vs.* 43.1 ± 11.7 | 5/30 *vs.* 6/29 | Balneotherapy exercises | 4/12 | Home self-exercises | BASDAI, BASFI, BASMI |
| Gunay, Keser & Bicer (2018) | RCT | 11 *vs.* 10 | 40 ± 11.40 *vs.* 44.5 ± 12.58 | NA | TENS + conventional exercises+balneotherapy | 3 | Conventional exercises | BASDAI, BASFI, BASMI, ASQoL |
| Gurcay et al. (2008) | RCT | 29 *vs.* 28 | 40.2 ± 10.38 *vs.* 41.3 ± 8.59 | 2/27 *vs.* 6/22 | Stager bath + conventional exercises | 3 | Conventional exercises | BASDAI, BASFI, BASMI, ASQoL |
| Karapolat et al. (2008) | RCT | 22 *vs.* 16 | 47.5 ± 11.78 *vs.* 46.6 ± 14.8 | 0/68 *vs.* 0/69 | Home self-exercises | 6 | Conventional exercises | BASDAI, BASFI, BASMI |
| Kjeken et al. (2013) | RCT | 46 *vs.* 49 | 49.4 ± 10.3 *vs.* 48.6 ± 9.4 | 10/36 *vs.* 23/25 | Muscular reinforcement +endurance training | 4/12 | Conventional exercises | BASDAI, BASFI |
| Lee et al. (2008) | nRCT | 13 vs17 | 35.2 ± 11.5 *vs.* 34.9 ± 12.9 | 10/3 *vs.* 15/2 | Tai chi | 8 | No intervention | BASDAI |
| Levitova et al. (2016) | nRCT | 22 | 36.78 ± 1.09 *vs.* 36.91 ± 1.16 | 4/18 | home-based exercise consisted of endurance | 24 | No intervention | BASDAI, BASFI, BASMI |
| Masiero et al. (2011) | RCT | 22 *vs.* 23 | 48.7 ± 18.0 *vs.* 46.9 ± 8.7 | 5/17 *vs.* 4/19 | rehabilitation plus an educational-behavioral balance, and strength training | 8/24 | Conventional exercises | BASDAI, BASFI, BASMI |
| Wang et al. (2022) | RCT | 26 *vs.* 28 | 31.2 ± 6.33 *vs.* 33.2 ± 6.2 | 6/20 *vs.* 7/21 | moderate-intensity Aerobic | 8/16 | Usual care | BASDAI, BASFI, BASMI |
| So et al. (2012) | RCT | 23 *vs.* 23 | 34.6 ± 5.9 *vs.* 38.0 ± 9.1 | 1/22 *vs.* 1/22 | Inspiratory muscle training, conventional exercises | 16 | usual care | BASDAI, BASFI |

Kong et al. (2025), *PeerJ*, DOI 10.7717/peerj.20336

**Table 1** (*continued*)

| Study | Participants (exercise *vs.* control) | | | | Interventions | | Comparator | Outcomes (BASDAI, BASFI, BASMI, ASQoL) |
|---|---|---|---|---|---|---|---|---|
| | Methodology | Sample | Age | Female/ | Type | Duration | | |
| Altan et al. (2012) | RCT | 25 *vs.* 30 | 46.5 ± 11.2 *vs.* 43.6 ± 10.1 | NA | Pilates | 12/24 | No intervention | BASDAI, BASFI, BASMI, ASQoL |
| Analay et al. (2003) | RCT | 23 *vs.* 22 | 37.6 ± 11.3 *vs.* 34.3 ± 7.9 | 3/20 *vs.* 4/18 | Aerobic exercises | 6/12 | Home self-exercises + educational therapy | BASFI |
| Aytekin et al. (2012) | nRCT | 34 *vs.* 32 | 34.35 ± 9.48 *vs.* 35.75 ± 6.71 | 9/25 *vs.* 5/27 | Home self-exercise (range of motion, stretching, strengthening, posture, and respiratory exercises) | 12 | No intervention | BASDAI, BASFI, ASQoL |
| Basakci Calik et al. (2021) | RCT | 17 *vs.* 14 | 46.58 ± 11.94 *vs.* 42.85 ± 11.07 | 9/8 *vs.* 10/4 | Aerobic supervised mobility exercises | 12/24 | Supervised mobility exercises | BASDAI, BASFI, BASMI |
| Ciprian et al. (2013) | RCT | 15 *vs.* 15 | 47.8 ± 10.0 *vs.* 45.6 ± 11.8 | 1/14 *vs.* 1/14 | spa therapy (mud packs and thermal baths) + rehabilitation (exercises in a thermal pool) + anti-TNF agents | 2/12/24 | receiving anti-TNF agents | BASDAI, BASMI |
| Günendi et al. (2010) | nRCT | 16 *vs.* 16 | 45.6 ± 12.4 *vs.* 43.4 ± 12.0 | 3/13 *vs.* 5/11 | Aerobic training + Home self-exercises (stretch and reinforcement) | 3 | Home self-exercises (stretch and reinforcement) | BASDAI, BASFI |
| Hsieh et al. (2014) | RCT | 9 *vs.* 10 | 36.2 ± 11.7 *vs.* 42.1 ± 8.8 | 3/6 *vs.* 3/7 | Aerobic training + muscular reinforcement + range-ofmotion home exercises HEP | 13 | Range-of-motion home exercises | BASDAI, BASFI |
| Jennings et al. (2015) | RCT | 35 *vs.* 35 | 42.9 ± 9.9 *vs.* 40.2 ± 9.3 | 9/26 *vs.* 12/23 | 50 min of walking + stretching exercises | 6/12/24 | stretching exercises | BASDAI, BASFI, BASMI |
| Karahan et al. (2016) | RCT | 28 *vs.* 29 | 36.1 ± 12.4 *vs.* 36.6 ± 11.3 | 6/24 *vs.* 7/23 | Home self-exercises (video-games) | 8 | No intervention | BASDAI, BASFI, ASQoL |
| Karapolat et al. (2009) | RCT | 12 *vs.* 12 *vs.* 13 | 50.15 ± 12.395 *vs.* 46.92 ± 13.399 *vs.* 48.42 ± 9.472 | 3/10 *vs.* 4/8 *vs.* 3/9 | (1) Swimming + conventional exercises (2) walking exercise +conventional exercises | 6 | Conventional exercises | BASDAI, BASFI, BASMI |
| Cagliyan et al. (2007) | RCT | 23 *vs.* 23 | 35.2 ± 7.8 *vs.* 36.8 ± 9.4 | 5/18 *vs.* 2/30 | Home self-exercises | 13 | No intervention | BASDAI, BASFI, BASMI, ASQoL |
| Masiero et al. (2014) | nRCT | 21 *vs.* 21 | 49.11 ± 11.8 *vs.* 46.15 ± 10.3 | 4/17 *vs.* 1/20 | Supervised exercises followed by self-exercises strength and balance | 48 | No intervention | BASDAI, BASFI, BASMI |
| Niedermann et al. (2013) | RCT | 53 *vs.* 53 | 50.1 ± 11.9 *vs.* 47.6 ± 12.4 | 19/34 *vs.* 19/34 | Aerobic training | 48 | Educational therapy | BASDAI, BASFI, BASMI |
| Rodriguez-Lozano et al. (2013) | RCT | 381 *vs.* 375 | 45 ± 12 *vs.* 46 ± 11 | 310/71 *vs.* 302/73 | Home self-exercises | 24 | No intervention | BASDAI, BASFI, ASQoL |
| Roşu et al. (2014) | RCT | 48 *vs.* 48 | 25.33 ± 3.77 *vs.* 24.98 ± 3.83 | 9/39 *vs.* 8/40 | Pilates, McKenzie and Heckscher McKenzie training (postural education, back stretching, pelvic stabilization | 48 | Conventional exercises | BASDAI, BASFI, BASMI |
| Rosu & Ancuta (2015) | RCT | 48 *vs.* 48 | 25.12 ± 3.98 *vs.* 22.96 ± 3.65 | 24/22 *vs.* 24/21 | Pilates, McKenzie and Heckscher | 48 | Conventional exercises | BASDAI, BASFI, BASMI |
| Silva, Andrade & Vilar (2012) | nRCT | 20 *vs.* 15 | NA | NA | Global postural rehabilitation | 16 | Conventional exercises | BASDAI |
| Sveaas et al. (2014) | RCT | 10 *vs.* 14 | 46.6 ± 13.6 *vs.* 49.9 ± 11.1 | 8/2 *vs.* 4/10 | Aerobic training (HiiT)+ muscular reinforcementGEP | 12 | No intervention | BASDAI, BASFI, BASMI |
| De Souza et al. (2017) | RCT | 27 *vs.* 28 | 45 ± 9.8 *vs.* 43.8 ± 10.2 | 7/23 *vs.* 9/21 | Swiss ball exercises | 8 | No intervention | BASDAI, BASFI, BASMI |
| Sveaas et al. (2020) | RCT | 50 *vs.* 50 | 45.8 ± 11.3 *vs.* 46.9 ± 11.3 | 25/25 *vs.* 28/22 | Aerobic training (HiiT)+muscular reinforcement | 12 | No intervention | BASDAI, BASFI, BASMI |
| Sweeney, Taylor & Calin (2002) | RCT | 100 *vs.* 100 | 47 ± 10.2 *vs.* 47 ± 9.6 | 30/70 *vs.* 32/68 | Home self-exercises | 24 | No intervention | BASDAI, BASFI |

Peerj

**Table 1** (*continued*)

| Study | Participants (exercise *vs.* control) | | | | Interventions | | Comparator | Outcomes (BASDAI, BASFI, BASMI, ASQoL) |
|---|---|---|---|---|---|---|---|---|
| | Methodology | Sample | Age | Female/ | Type | Duration | | |
| *Xie et al. (2019)* | RCT | 23 *vs.* 23 | NA | 6/17 *vs.* 5/18 | Baduajin Qigong | 12 | No intervention | BASDAI, BASFI, BASMI |
| *Acharya et al. (2024)* | RCT | 15 *vs.* 18 | 32.2 ± 5.72 *vs.* 31.6 ± 6.19 | 2/13 *vs.* 1/17 | Yoga | 8/12 | Conventional exercises | BASFI, BASMI |
| *So et al. (2012)* | RCT | 23 *vs.* 23 | 34.6 ± 5.9 *vs.* 38.0 ± 9.1 | 1/22 *vs.* 1/22 | incentive spirometer exercise + Conventional exercises | 16 | Conventional exercises | BASDAI, BASFI |
| *Durmus et al. (2009)* | RCT | 25 *vs.* 18 | 37.34 ± 7.33 *vs.* 42.32 ± 8.19 | 4/21 *vs.* 4/14 | home-based exercise | 12 | medical therapy | BASDAI, BASFI |
| *Widberg, Karimi & Hafström (2009)* | RCT | 16 *vs.* 16 | 40.5 ± 7.8 *vs.* 36.5 ± 7.5 | NA | HEP (mobility exercises, stretching and relaxation exercises) | 8 | control group | BASDAI, BASFI, BASMI |

**Notes.**

NA, not available; BASDAI, Bath Ankylosing Spondylitis Disease Activity Index; BASFI, Bath Ankylosing Spondylitis Functional Index; BASMI, Bath Ankylosing Spondylitis Metrology Index; ASQoL, Ankylosing Spondylitis Quality of Life.

For all the outcomes, a lower score means better function.

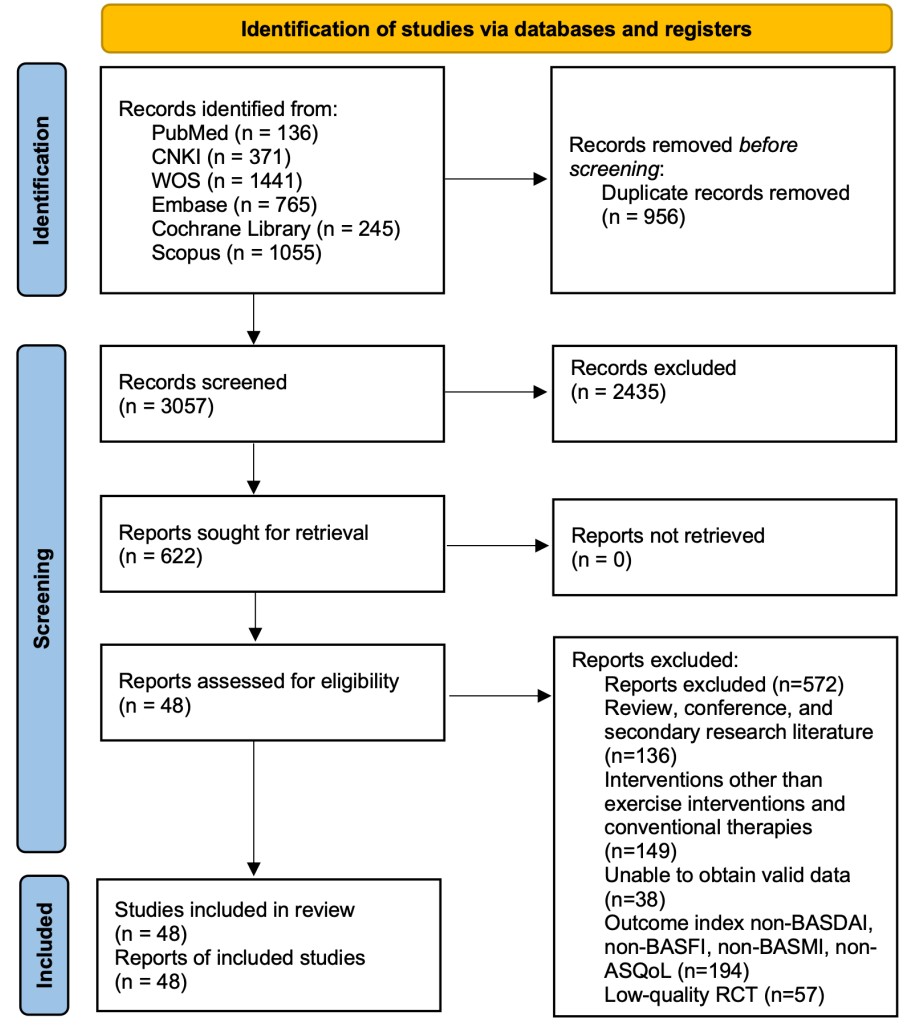

**Figure 1  PRISMA 2020 flow diagram.**

between CG and LSE and CG and LAE. The CHC intervention appears less studied, as shown by its smaller node and limited direct comparisons. Despite this, the network remains robust, making it a suitable framework for a network meta-analysis. Figure 2C presents the network structure for interventions targeting BASMI. Here, LAE, LSE, and CG emerge as the most extensively studied interventions, reflected in their larger node sizes. The strongest direct comparison was between LAE and CG, reaffirming their status as key intervention pairs. The network structure in this diagram highlights the significant interventions and their relationships, providing valuable insights for subsequent effect ranking analyses. Figure 2D offers a more streamlined representation, focusing on exercise interventions in relation to ASQoL. The diagram features fewer nodes, centering on CG, LAE, AAE, and ASE. The most substantial comparison between CG and LSE suggests that there should be greater research attention on this pair. While this network can assess the

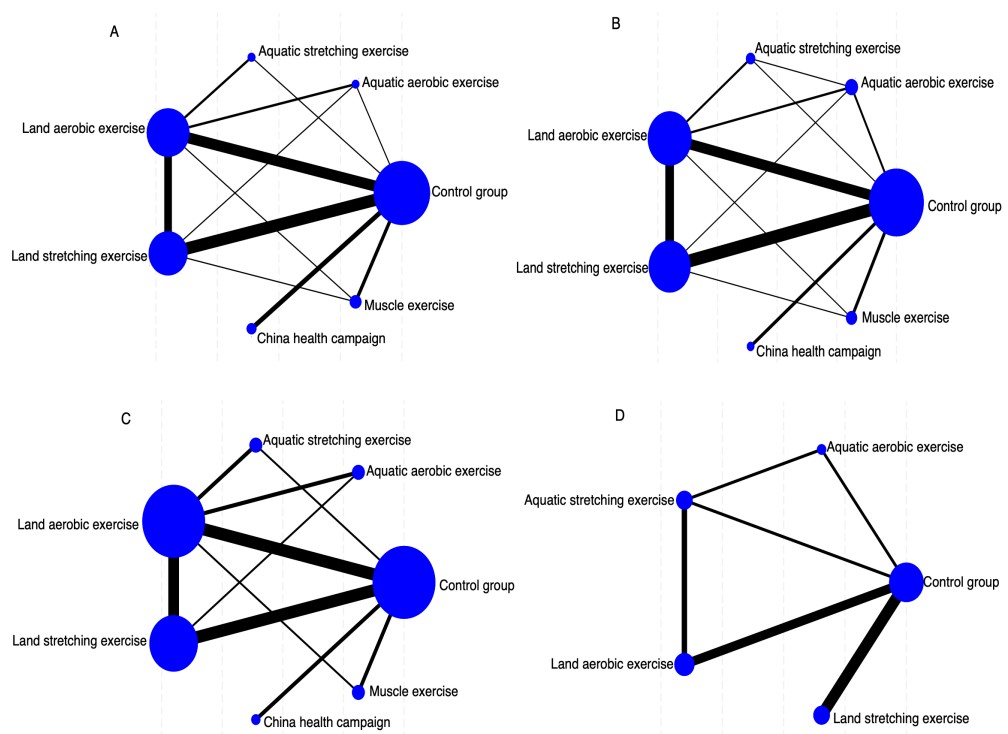

**Figure 2** Network geometry indicates several participants in each arm (size of points) and comparisons between arms (thickness of lines). (A) BASDAI, (B) BASFI, (C) BASMI, and (D) ASQoL.

relative effects of different exercise interventions, the structure is comparatively sparse, limiting the availability of indirect comparisons.

## Risk of bias within studies

Figure 3 summarizes the risk of bias assessment for each study. Blinding of participants and staff was rarely reported, leading to a generally high risk of bias across most studies. This could introduce potential biases, such as expectation effects, affecting effect size estimations. Given the high risk of bias in the comparator studies, the certainty rating of all treatment groups was downgraded accordingly.

## Main outcome: BASDAI

For the BASDAI index, our network meta-analysis incorporated 38 randomized controlled trials (RCTs) and four non-randomized controlled trials (nRCTs), encompassing 2,852 patients with ankylosing spondylitis. The distribution of studies across different exercise interventions was as follows: AAE (3), ASE (3), LAE (21), LSE (19), CHC (5), and ME (5). Our findings indicate that all exercise interventions outperformed the control group in improving BASDAI, as illustrated in Fig. 4A. More specifically, ASE [−1.42, 95% CI [−2.51 to −0.33], LAE [−0.94, 95% CI [−1.41 to −0.47]], and LSE [−0.49, 95% CI [−0.94 to −0.04]] demonstrated statistically significant benefits over conventional treatment (CG) in reducing ankylosing spondylitis symptoms. Meanwhile, AAE [−1.13,

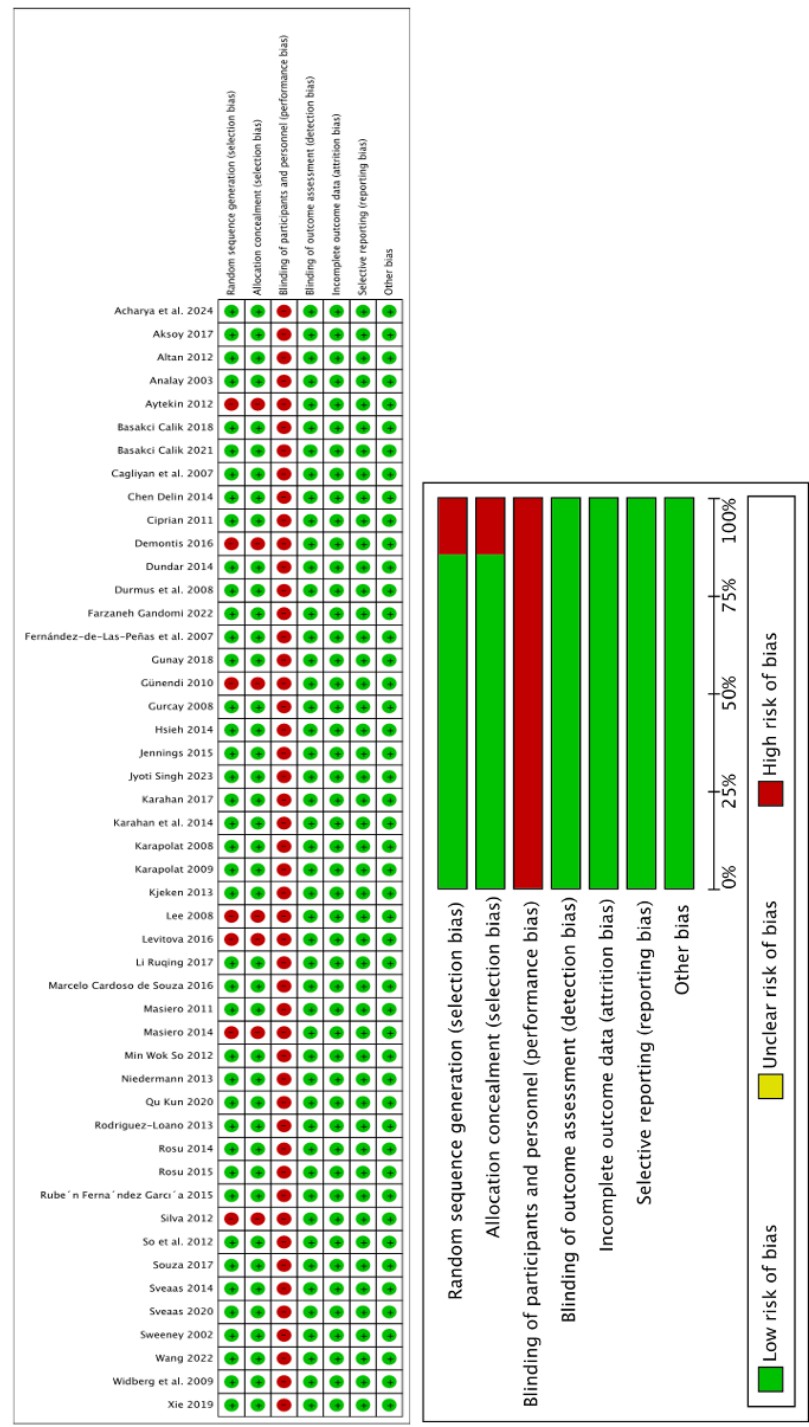

**Figure 3  Risk of bias summary.**

95% CI [−2.34 to 0.08]], CHC [−0.55, 95% CI [−1.45 to 0.35]], and ME [−0.73, 95% CI [−1.49 to 0.03]] showed some potential for improvement, though their effects were not statistically significant. No significant differences were detected between exercise

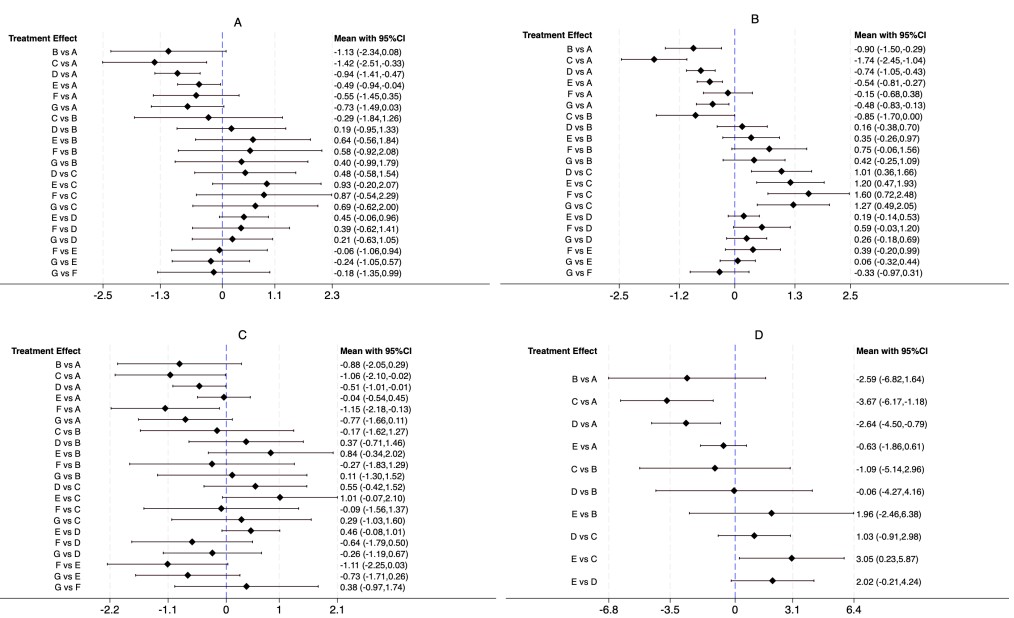

**Figure 4  Direct and indirect comparison between interventions.** (A) BASDAI, (B) BASFI, (C) BASMI, and (D) ASQoL.

interventions in indirect comparisons. To determine the most effective intervention, we utilized the SUCRA method, a widely adopted ranking approach in network meta-analysis that accounts for both effect estimates and their uncertainty (*Chen, Zhang & Zhu, 2023*). Figure 5A and Table 2 reveal that ASE had the highest SUCRA score (85.5), followed by AAE (71.3), LAE (66.5), ME (51.2), CHC (39.6), and LSE (32.6), with the control group scoring the lowest (3.4). These results suggest that ASE is the most effective intervention for improving BASDAI.

## Main outcome: BASFI

For the BASFI assessment, our network meta-analysis included 38 randomized controlled trials (RCTs) and five non-randomized controlled trials (nRCTs), with a total of 2,924 participants. The studies were distributed across the following interventions: AAE (four), ASE (three), LAE (22), LSE (21), CHC (three), and ME (five). Results from the network meta-analysis demonstrated that all exercise interventions outperformed the CG in improving BASFI, as shown in Fig. 4B. Further analysis revealed that AAE [−0.90, 95% CI [−1.50 to −0.29]], ASE [−1.74, 95% CI [−2.45 to −1.04]], LAE [−0.74, 95% CI [−1.05 to −0.43]], LSE [−0.54, 95% CI [−0.81 to −0.27]], and ME [−0.48, 95% CI [−0.83 to −0.13]] demonstrated statistically significant improvements compared to CG, whereas CHC [−0.15, 95% CI [−0.68 to 0.38]] did not show a significant effect. Indirect comparisons indicated that ASE significantly outperformed AAE [−0.85, 95% CI [−1.70 to 0.00]], whereas no significant differences were found between AAE and LAE, LSE, CHC, or ME. When compared to ASE, however, LAE, LSE, CHC, and ME all showed

Kong et al. (2025), *PeerJ*, DOI 10.7717/peerj.20336

**Table 2  SUCRA, PrBest, and MeanRank for each intervention.**

| Exercise | BASDAI | | | BASFI | | | BASMI | | | ASQoL | | |
|---|---|---|---|---|---|---|---|---|---|---|---|---|
| | SUCRA | PrBest/% | MeanRank | SUCRA | PrBest/% | MeanRank | SUCRA | PrBest/% | MeanRank | SUCRA | PrBest/% | MeanRank |
| CG | 3.4 | 0.0 | 6.8 | 4.9 | 0.0 | 6.7 | 9.9 | 0.0 | 6.4 | 7.2 | 0.0 | 4.7 |
| AAE | 71.3 | 29.9 | 2.7 | 74.4 | 2.2 | 2.5 | 65.9 | 20.5 | 3.0 | 62.4 | 29.1 | 2.5 |
| ASE | 85.5 | 55.4 | 1.9 | 99.6 | 97.8 | 1.1 | 75.6 | 30.7 | 2.5 | 88.4 | 60.4 | 1.5 |
| LAE | 66.5 | 4.9 | 3.0 | 66.9 | 0.0 | 3.0 | 45.4 | 0.1 | 4.3 | 65.3 | 10.1 | 2.4 |
| LSE | 32.6 | 0.1 | 5.0 | 46.5 | 0.0 | 4.2 | 14.0 | 0.0 | 6.2 | 26.7 | 0.4 | 3.9 |
| CHC | 39.6 | 4.2 | 4.6 | 17.5 | 0.0 | 5.9 | 78.7 | 38.4 | 2.3 | / | / | / |
| **ME** | 51.2 | 5.5 | 3.9 | 40.2 | 0.0 | 4.6 | 60.6 | 10.3 | 3.4 | / | / | / |

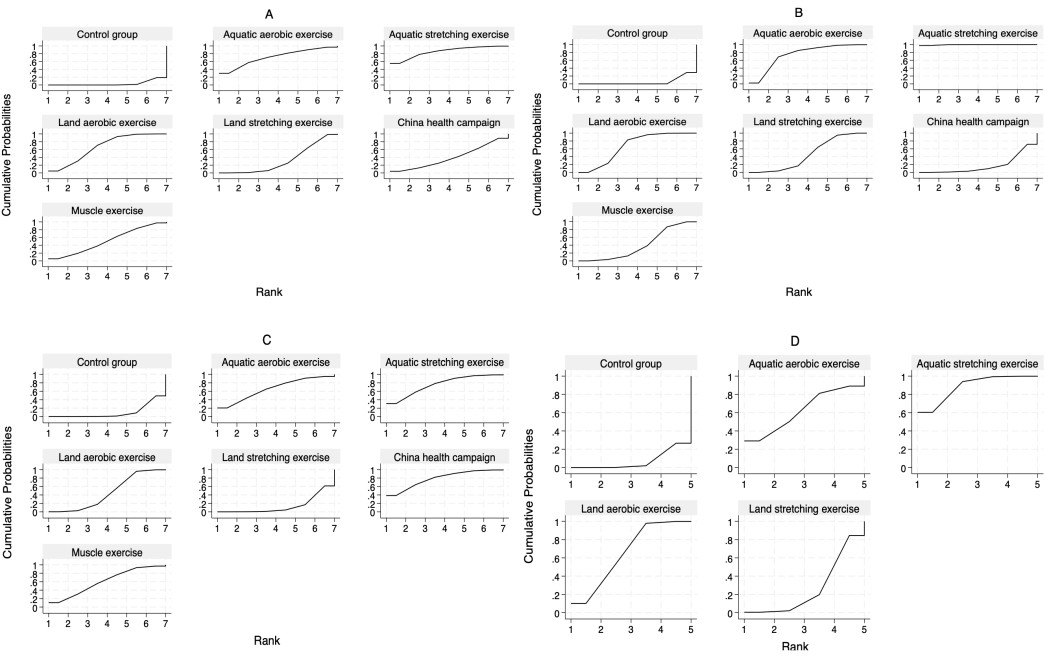

**Figure 5** **SUCRA curves of the effectiveness of each exercise form.** The larger the area under the SU-CRA curve, the greater the cumulative probability of ranking high for that intervention. (A) BASDAI, (B) BASFI, (C) BASMI, and (D) ASQoL.

statistically significant differences, suggesting that ASE was a more effective intervention. No significant differences were observed when comparing LAE with LSE, CHC, or ME, and similar results were found in pairwise comparisons between LSE, CHC, and ME. According to the SUCRA ranking results (Fig. 5B, Table 2), ASE ranked the highest (99.6), followed by AAE (74.4), LAE (66.9), LSE (46.5), ME (40.2), CHC (17.5), and CG (4.9). These findings further confirm that ASE is the most effective intervention for improving BASFI.

## Main outcome: BASMI

For the BASMI assessment, 25 randomized controlled trials (RCTs) and three non-randomized controlled trials (nRCTs) involving 1,459 participants were included. The number of studies for each intervention was as follows: AAE (two), ASE (three), LAE (16), LSE (12), CHC (three), and ME (three). As shown in Fig. 4C, the results of the network meta-analysis showed that compared to the CG, ASE [−1.06, 95% CI [−2.10 to −0.02]], LAE [−0.51, 95% CI [−1.01 to −0.01]], and CHC [−1.15, 95% CI [−2.18 to −0.13]] exhibited statistically significant improvements. However, while AAE [−0.88, 95% CI [−2.05 to 0.29]], LSE [−0.04, 95% CI [−0.54 to 0.45]], and ME [−0.77, 95% CI [−1.66 to 0.11]] showed a trend toward improvement, their effects did not reach statistical significance. Additionally, no significant differences were observed in indirect comparisons. The SUCRA ranking results (Fig. 5C, Table 2) indicated that CHC had the highest SUCRA score (78.7), followed by ASE (75.6), AAE (65.9), ME (60.6), LAE (45.4),

LSE (14.0), and CG (9.9). These findings suggest that CHC holds the greatest potential for improving BASMI.

### Main outcome: ASQoL

For the ASQoL assessment, a total of nine randomized controlled trials (RCTs) and one non-randomized controlled trial (nRCT) were included, involving 1,165 participants. The number of studies for each intervention was as follows: AAE (one), ASE (three), LAE (five), and LSE (four). As illustrated in Fig. 4D, the results of the network meta-analysis indicated that compared to the CG, ASE [−3.67, 95% CI [−6.17 to −1.18]] and LAE [−2.64, 95% CI [−4.50 to −0.79]] demonstrated statistically significant advantages. Although AAE [−2.59, 95% CI [−6.82 to 1.64]] and LSE [−0.63, 95% CI [−1.86 to 0.61]] showed a trend toward improvement, their effects did not reach statistical significance. Indirect comparisons revealed a significant difference between LSE and ASE [3.05, 95% CI [0.23–5.87]], while no other significant differences were observed. The SUCRA ranking results (Fig. 5D, Table 2) showed that ASE ranked highest (88.4), followed by LAE (65.3), AAE (62.4), LSE (26.7), and CG (7.2), further supporting its superior efficacy in improving quality of life.

### Heterogeneity and inconsistency assessment

To ensure the reliability of our findings, we evaluated the inconsistency across all outcome measures. As shown in Appendix S4, the *p*-values for global, node, and loop inconsistencies in BASDAI, BASFI, BASMI, and ASQoL were all above 0.05, suggesting that the results were consistent and free from significant inconsistencies.

### Publication bias

Because forty or more studies were included in the meta-analysis of the effects of exercise interventions on BASDAI (disease activity), BASFI (functional impairment), BASMI (spinal mobility), and ASQoL (quality of life), we used a funnel plot to test their publication bias. Appendix S5 shows that the scatter is upward distributed, with a fundamental balance between left and right and no significant publication bias among the studies.

## DISCUSSION

This study utilized a network meta-analysis to systematically compare the effectiveness of different exercise interventions on four key outcomes in AS patients: BASDAI, BASFI, BASMI, and ASQoL. A total of 48 studies, covering 3,140 participants, were analyzed. While a previous meta-analysis had assessed the impact of exercise on AS symptoms, it included only 10 trials, offered limited exercise variety, and lacked a systematic comparison of different exercise types. Notably, that study relied on just one source for interventions such as Exergame, Swiss ball training, Pilates, yoga, and running (*Luo et al., 2024*), limiting its findings' breadth and depth. In contrast, our study offers the first comprehensive comparison of six major exercise types, significantly broadening the scope of previous research. The results showed that all exercise interventions outperformed non-exercise controls. Among them, ASE ranked highest across three key measures—BASDAI (SUCRA = 85.5), BASFI (SUCRA = 99.6), and ASQoL (SUCRA = 88.4)—indicating its broad and
significant benefits. CHC was the most effective in improving BASMI (SUCRA = 78.7). Other exercise types, such as AAE, LAE, and ME, also provided some benefits, though ASE consistently showed the strongest overall effect.

For BASDAI, which reflects disease activity, 42 studies (38 RCTs and four nRCTs) were analyzed. The findings confirmed that all exercise interventions, except for the control group, helped reduce disease activity to varying degrees. Among them, ASE was the most effective (SUCRA = 85.5) and ranked higher than other interventions. This aligns with previous research but offers more detailed insights into the relative effectiveness of different exercise types. Prior studies often focused on single exercise modalities. For example, *Boudjani et al. (2023)* found that regular exercise reduced pain and stiffness in AS patients. Still, their analysis was limited to low-intensity stretching and aquatic exercises without comparing different exercise types. Similarly, *Luo et al. (2024)* attempted to compare different exercises but only included 10 studies with small sample sizes and did not assess multi-component interventions like ASE. Our study provides more substantial clinical guidance by including a larger body of research ($n = 42$) and ranking the relative effectiveness of various interventions. The superior efficacy of ASE is likely due to its combination of aerobic and resistance training. Aerobic exercise improves cardiovascular health and modulates inflammatory markers like TNF-$\alpha$ and IL-6 (*Chen et al., 2025*), while resistance training helps prevent muscle loss, enhances postural control, and directly addresses key BASDAI symptoms such as morning stiffness, fatigue, and joint pain (*Acar, Ilçin & Sarı, 2023*). While LAE and LSE also showed significant benefits, their rankings (SUCRA = 66.5 and 32.6, respectively) were lower than ASE, suggesting that single-modality exercises may be less effective in addressing the complex pathology of AS. AAE, CHC, and ME showed trends toward improvement, but their confidence intervals included zero, indicating that current evidence is insufficient to confirm their effects on disease activity. This could be due to the fewer included studies or inconsistencies in exercise frequency and intensity. It is worth noting that although ASE ranked first, only three studies assessed it, highlighting the need for more high-quality RCTs to validate its effectiveness. Nonetheless, the available data suggest that ASE holds significant potential as a comprehensive intervention for AS and merits further clinical application.

In terms of BASFI, which measures functional ability, nearly all exercise interventions showed significant improvements compared to the control group, with ASE being the most effective (SUCRA = 99.6). This supports our earlier findings in the BASDAI analysis, suggesting that multi-component exercise programs may provide the most well-rounded benefits for AS patients. Functional impairment in AS often leads to limited spinal movement and difficulties in daily activities. ASE integrates aquatic stretching, which enhances muscle strength and joint flexibility, improves circulation, and reduces inflammation, helping patients regain mobility. Other exercise types, including AAE, LAE, LSE, and ME, also demonstrated statistically significant improvements, consistent with prior research. For instance, *Azab et al. (2022)* found that land-based aerobic exercise significantly improved gait and postural control in AS patients, promoting independence in daily activities. Resistance training has also been shown to strengthen core muscles, enhance spinal stability, and improve posture (*Hlaing et al., 2021*). However, CHC did not

show significant improvements in BASFI, likely due to limited study numbers and high variability in intervention methods. This suggests that CHC may be more effective for specific outcomes rather than general functional improvement.

For BASMI, which assesses spinal mobility, CHC emerged as the top-ranking intervention (SUCRA = 78.7), outperforming ASE and AAE. This suggests that CHC-based interventions, which include Tai Chi, Baduanjin, and Qigong, may be particularly beneficial for patients with severe spinal stiffness and pain. These exercises emphasize controlled, adaptive movements, which can be especially helpful for AS patients with limited mobility. Additionally, the buoyancy provided by aquatic exercises reduces spinal load, facilitates joint movement, and gradually improves flexibility (*Ansari, Elmieh & Alipour, 2021*). This finding fills a research gap regarding the effects of CHC and highlights its potential role in improving spinal function. ASE ranked second in BASMI (SUCRA = 75.6), further reinforcing its broad applicability across multiple outcome measures. While AAE and ME did not reach statistical significance, their relatively high SUCRA rankings suggest potential benefits that warrant further study.

For ASQoL, ASE again ranked highest (SUCRA = 88.4), showing clear advantages over LAE and particularly over LSE [3.05, 95% CI [0.23–5.87]]. This suggests that ASE provides more comprehensive benefits for both physical and mental well-being than low-intensity single-modality exercises. Quality of life in AS is influenced by symptom relief and psychological well-being, self-efficacy, and social participation. Combined aerobic and resistance training can enhance patients' confidence and motivation to engage in physical activity, ultimately leading to improved quality of life—an effect well-documented in chronic disease management research (*Fiorilli et al., 2022*). Although AAE and LSE did not reach statistical significance in ASQoL, their improvement trends suggest they may still be clinically relevant. Some studies' limited sample sizes and short intervention durations may have underestimated their effectiveness. Future research should explore how different exercise types specifically impact various aspects of quality of life.

Despite its strengths, our network meta-analysis has several limitations. First, while a considerable number of studies were included, some exercise modalities (*e.g.*, AAE, ASE, and CHC) were assessed in a limited number of trials, which may have compromised the reliability and precision of their estimated effects. Second, methodological weaknesses in some studies—such as unclear randomization processes and blinding—introduce a risk of bias. Third, substantial variation in intervention duration, frequency, and intensity across studies may have influenced treatment effects and restricted our ability to establish a definitive "optimal exercise prescription." Additionally, specific outcomes, such as ASQoL, were evaluated in relatively few studies, and the stability of SUCRA rankings requires further validation in larger-scale research. Lastly, our study primarily relied on English and Chinese databases, which may have led to the omission of relevant literature and introduced potential publication bias.

## CONCLUSIONS

This study utilized a network meta-analysis to quantify the comparative effectiveness of various exercise interventions for AS, offering clinical guidance for optimizing exercise-based treatment plans. The findings suggest that ASE is the most effective intervention for improving disease activity (BASDAI), functional ability (BASFI), and quality of life (ASQoL). At the same time, CHC provides the most significant benefits for spinal mobility (BASMI). Given the variability in treatment responses among different exercise modalities, the study underscores the importance of personalized exercise prescriptions tailored to individual patient needs. Future research should focus on high-quality studies to further clarify these interventions' long-term benefits and mechanisms, promoting the widespread application of evidence-based rehabilitation strategies in AS management.

### Funding

This research was supported by the Heze University 2025 Doctoral Fund (Grant No. XY25BS02).

### Grant Disclosures

The following grant information was disclosed by the authors:
Heze University: XY25BS02.

### Competing Interests

The authors declare there are no competing interests.

### Author Contributions

- Lingkui Kong conceived and designed the experiments, performed the experiments, analyzed the data, prepared figures and/or tables, and approved the final draft.
- Chuanwen Yu conceived and designed the experiments, performed the experiments, analyzed the data, prepared figures and/or tables, authored or reviewed drafts of the article, and approved the final draft.
- Chaoxin Wang conceived and designed the experiments, performed the experiments, authored or reviewed drafts of the article, and approved the final draft.
- Zhanpeng Meng analyzed the data, prepared figures and/or tables, and approved the final draft.

### Data Availability

The raw measurements are available in the Supplementary File.

### Supplemental Information

Supplemental information for this article can be found online at http://dx.doi.org/10.7717/peerj.20336#supplemental-information.

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
