# Peer review of "Comparative efficacy of different exercise interventions in patients with ankylosing spondylitis: a systematic review and network meta-analysis"

_PeerJ, doi:10.7717/peerj.20336_

## Round 0.1 · original submission · Major Revisions

· Academic Editor

Major Revisions

**Language Note:** When you prepare your next revision, please either (i) have a colleague who is proficient in English and familiar with the subject matter review your manuscript, or (ii) contact a professional editing service to review your manuscript. PeerJ can provide language editing services - you can contact us at [email protected] for pricing (be sure to provide your manuscript number and title). – PeerJ Staff

·

Basic reporting

In this meta-analysis, the authors performed a systematic review of the effect of different exercise approaches on people with Ankylosing spondylitis (AS) in the literature that provides a quantitative estimate of the impact of exercise training. The author used the techniques of a structured and standardized approach for analyzing prior findings in exercise therapy in the literature. As a reader, I found enough information to determine whether the included studies were appropriate for a combined analysis. I thoroughly enjoyed reviewing this manuscript and only have some minor requests for revision. Please find below my further evaluation and suggestions.

Experimental design

> Aim:
-The authors should state the aims clearly and clinically in the introduction. Relevant and focused study questions should be included.
- In the introduction, it is useful to briefly mention any previous meta-analyses conducted for similar purposes, along with their results (please check DOI: 10.1080/09638288.2022.2140842 and DOI: 10.1002/msc.1641). It is also helpful to emphasize how this meta-analysis contributes to the subject. These differences can also be used in the discussion.

> - Literature search:
A comprehensive literature search was conducted. Searched information sources listed (ie, PubMed, Scopus, Cochrane Library, and EBSCO). Terms used for electronic literature search are provided. Reasonable limitations placed on search (ie, English and Chinese language)
- Exclusion criteria: The exclusion of studies using hippotherapy simulation in treatment should be indicated in this section.

- During my literature search, I noticed that one particularly unique article had been omitted (please see the following DOI: 10.1080/07853890.2023.2249822). This article reports the first use of hippotherapy simulators in the treatment of ankylosing spondylitis. If you include this study in your meta-analysis, I believe it will make a valuable contribution to your article. Additionally, the fact that this article is new and has not been included in previous meta-analyses will enhance your article.

> -Data abstraction:
> Structured data abstraction form used. Number of authors (>2) who abstracted data given. Characteristics of studies listed (ie, sample size, patient demographics). Inclusion and exclusion criteria were provided for studies. The number of excluded studies and reasons for exclusion are included.

Validity of the findings

Evaluation of results:
Appropriate statistical methods were used to combine results.
Results were displayed. Sensitivity analyses were conducted.

Discussion section
-The results were discussed from different perspectives. They were put into context without over-interpretation. The conclusions were in line with the aims and objectives of the study.

References:
- The references meet the standards set out in the journal's editorial guidelines.

Reviewer 2 ·

Basic reporting

The manuscript is written in professional, clear English with appropriate terminology throughout. The authors define the aim of the study.

The article includes a sufficient introduction and provides adequate background, emphasizing the importance of exercise in Ankylosing Spondylitis (AS). The rationale for performing a network meta-analysis (NMA) is justified by the diversity of exercise options and the lack of head-to-head trials.
The manuscript adheres to the PRISMA guideline. The PRISMA flow diagram, network geometry, and tables are informative. Raw data is shared. Supplementary materials are available.

References
Several references cited in the manuscript do not accurately reflect the original sources, which may lead to misleading interpretations. For example;
1) ASAS-EULAR 2022 guideline: The authors cited the study by Kone-Paut et al. (2021) when stating that exercise is considered a core intervention in the treatment of AS. However, this citation does not appear to be the 2022 ASAS-EULAR guideline publication.
2) Stata 18.0 software: The Stata 18.0 software used in the analyses was referenced by Shim et al. (2017). This study is not directly related to the software. The correct reference should be the official source published by StataCorp.
3) Cochrane Handbook citation: The Vrabel (2015) study was used for citation purposes in the Cochrane Handbook. However, the Cochrane Handbook should generally be cited through the versions published under the editorship of Higgins et al.
4) Strength training: This study has been used to support the claim that strength training reduces structural damage. However, the study is protocol-based and does not yet offer clinical results.
5) Swiss ball: The effects of Swiss ball, Pilates, and flexibility training on spinal extension and quality of life were referenced. However, the study only examined strength training with a Swiss ball; Pilates or general flexibility training were not addressed.

These issues suggest the need for a thorough revision of the manuscript’s reference accuracy to ensure each citation faithfully represents the source material.

Experimental design

The study is based on a well-defined and clinically relevant research question. A network meta-analysis approach was adopted to compare various exercise interventions in patients with AS, aiming to fill a significant gap in the literature. Overall, the design demonstrates a significant effort to synthesize existing evidence. However, some methodological improvements are needed to increase transparency and reproducibility.

In particular, the classification of exercise types (e.g., ASE, CHC, ME) and the rationale underlying this classification are not clearly presented. This may limit the interpretability of subgroup comparisons. Furthermore, it is unclear how differences in factors such as exercise frequency, intensity, and duration were addressed across the included studies, raising questions about intervention heterogeneity.

Furthermore, the sources of some references used in the study were not accurately cited. More importantly, some sources were used to support claims that exceeded their content. Reference inconsistencies and misrepresentations can undermine the credibility of the synthesized findings and should therefore be reviewed and corrected.

Validity of the findings

This study provides an important and comprehensive analysis comparing the effects of different types of exercise in patients with AS. The findings are generally consistent, and meaningful comparisons were made using network meta-analysis. However, the fact that some sources do not support the findings as stated in the study requires caution when interpreting the results. For example, some references appear to support these interventions despite not directly examining the specific exercise types or outcomes. This could undermine the validity of conclusions suggesting that certain types of exercise are more effective than others. Therefore, to enhance the scientific robustness of the study, it is beneficial to use references more accurately and carefully. A more cautious interpretation of the findings will allow the reader to interpret the results with greater confidence.

Additional comments

This study is a comprehensive network meta-analysis evaluating the comparative effectiveness of different types of exercise in patients with ankylosing spondylitis and has the potential to be a valuable contribution to the field. However, the use of some references incorrectly or out of context undermines the scientific validity of statements about the superiority of certain types of exercise. In particular, findings from some studies were presented in an overly broad or compelling manner, and some references were cited as supporting interventions that they had not examined at all. In addition, citing incorrect sources in the methodology section negatively affects the reliability of the study.

---

## Round 0.2 · Minor Revisions

· Academic Editor

Minor Revisions

·

Basic reporting

The article provides an adequate introduction and background to show how the work fits into the wider area of knowledge. The text includes appropriate references to relevant prior literature.

Experimental design

The research question is clearly defined in the submission. Researchers have identified a knowledge gap in the literature and made statements about how the study contributes to filling that gap.

Validity of the findings

The results section contains robust, statistically sound, and controlled data.

Additional comments

I am satisfied with the author’s responses to questions/issues raised in the initial review. The authors responded adequately to each of the questions and made a few changes to reflect their responses in the manuscript itself. The revised manuscript is easier to follow based on feedback from the reviewers.

Overall;
The study design is suitable for answering the aim.
The article is consistent within itself.
The manuscript is ready for publication as it is.

Reviewer 2 ·

Basic reporting

I appreciate the authors’ thorough revisions and careful attention to the previous comments. The manuscript has significantly improved in clarity and overall quality.

Experimental design

I have only a few additional suggestions for refinement:

1. Page 3, line 12–13: The sentence “On the molecular level, physical activity has been shown to modulate the IL-17/IL-23 axis and suppress inflammatory responses (Landgren et al., 2023)” does not appear to be directly supported by the cited reference. I recommend replacing it with a more appropriate source that specifically addresses the modulation of the IL-17/IL-23 axis by physical activity.

2. Page 3, line 16–17: The statement “…while at the biomechanical level, it improves paraspinal muscle stability and slows disease progression (Wang et al., 2024)” is not fully aligned with the cited reference. The systematic review and meta-analysis by Wang et al. (2024) primarily reports improvements in physical function, disease activity, pain, spinal mobility, and quality of life among axSpA patients, but does not explicitly discuss paraspinal muscle stability or disease progression. I suggest revising this sentence for accuracy or providing a more suitable reference.

3. Introduction: To further strengthen the rationale for the present study, I recommend briefly acknowledging observational evidence indicating that higher physical activity levels are associated with better physical function, reduced disease activity, enhanced quality of life, and favorable psychological outcomes in AS. Adding a brief mention of this evidence after discussing the limitations of systematic reviews would provide a stronger context and emphasize the clinical relevance of the current work.

Validity of the findings

I appreciate the authors’ thorough revisions and careful attention to the previous comments.

Additional comments

Overall, the manuscript is much improved, and I believe addressing these minor issues will further enhance its accuracy.

---

## Round 0.3 · accepted · Accept

· Academic Editor

Accept

The authors have addressed all of the reviewers' comments. The manuscript is ready for publication.

Reviewer 2 ·

Basic reporting

I appreciate the authors' thorough revisions and careful attention to the previous comments. The manuscript has significantly improved in clarity and overall quality.

Experimental design

I am pleased with the authors' responses to the reviewer comments. They have addressed each point with care, and their revisions have significantly strengthened the manuscript. Overall, the updated version demonstrates notable improvements in clarity, structure, and depth, making it much more cohesive and accessible.

Validity of the findings

I appreciate the authors' thorough revisions and careful attention to the previous comments.

Additional comments

In conclusion, the revised manuscript reflects significant enhancements in clarity, structural coherence, and scholarly depth, resulting in a more polished and accessible contribution to the field.